# Clinicians’ and Users’ Views and Experiences of a Tele-Mental Health Service Implemented Alongside the Public Mental Health System during the COVID-19 Pandemic

**DOI:** 10.3390/ijerph20105870

**Published:** 2023-05-19

**Authors:** Anton Isaacs, Eleanor K. L. Mitchell, Keith Sutton, Michael Naughton, Rochelle Hine, Shane Bullock, Denise Azar, Darryl Maybery

**Affiliations:** 1School of Rural Health, Monash University, Warragul, VIC 3820, Australia; eleanor.mitchell@monash.edu (E.K.L.M.); keith.sutton@monash.edu (K.S.); rochelle.hine@monash.edu (R.H.); shane.bullock@monash.edu (S.B.); darryl.maybery@monash.edu (D.M.); 2Gippsland Primary Health Network, Traralgon, VIC 3844, Australia

**Keywords:** tele-mental health, telemedicine, COVID-19, mental health services, telehealth, referral and consultation, treatment, qualitative research

## Abstract

A tele-mental health model called Head to Health was implemented in the state of Victoria, Australia to address the crisis caused by the COVID-19 pandemic. It was a free centralized intake service that adopted a targeted approach with several novel elements, such as stepped care and telehealth. This study examines the views and experiences of clinicians and service users of the tele-mental health service in the Gippsland region of Victoria during the COVID-19 pandemic. Data from clinicians were obtained via an online 10-item open-ended survey instrument and from service users through semi-structured interviews. Data were obtained from 66 participants, including 47 clinician surveys and 19 service user interviews. Six categories emerged from the data. They were: ‘Conditions where use of tele-mental health is appropriate’, ‘Conditions where tele-mental health may not be useful’, ‘Advantages of tele-mental health’, ‘Challenges in using tele-mental health’, ‘Client outcomes with tele-mental health’, and ‘Recommendations for future use’. This is one of a few studies where clinicians’ and service users’ views and experiences have been explored together to provide a nuanced understanding of perspectives on the efficacy of tele-mental health when it was implemented alongside public mental health services.

## 1. Introduction

According to the Australian Government’s Department of Health, telehealth is the ‘use of telecommunication techniques for the purpose of providing telemedicine, medical education, and health education over a distance’ [1]. Tele-mental health (TMH) refers to ‘the use of information and communications technologies, including videoconferencing, to deliver mental health care remotely, including evaluations, medication management, and psychotherapy’ [2,3]. While TMH has been found to be an effective and acceptable form of service delivery [4], it also has some shortcomings.

For instance, the lack of internet access, the lack of computers or computer software [5], and preference for face-to-face interactions [6] are barriers that have been reported previously. Tele-mental interventions do not yet have proven effectiveness when used with children and young people with mental health problems [7,8], and in older persons, prior experience with technology is a critical factor [9]. In addition, when used in place of face-to-face mental health services, not being present with the patient experiencing a crisis has been suggested as a significant and potentially precarious situation [5]. Further evidence is needed to determine what types of interventions are effective, for which patients and in which settings, and whether such interventions can be used as a replacement for the standard treatment [10]. In addition, whether or not tele-mental technologies are associated with any harm also needs investigation.

Mental health services in the state of Victoria in Australia’s southeast have been reported to be ‘broken’ and a large group of individuals have been left with unmet needs for services [11]. The term ‘missing middle’ is often used to refer to a group of individuals who are too unwell for primary care but not unwell enough for specialized mental health services [12]. This was one of many service gaps that resulted in a Royal Commission into Victoria’s Mental Health System, which, among its many recommendations, stated that the system needed to be adaptive so that it ‘can identify and test new ideas, gather evidence about what works, and translate this into effective treatment, care and support’ [13].

At about this time, the COVID-19 pandemic swept across the world. The state of Victoria went into a 111-day lockdown which resulted in a surge in the number of people needing urgent mental health care. Several countries turned to TMH in the context of lockdown restrictions as there was an urgent need to address the sudden increase in mental health problems in the population [14]. The state of Victoria implemented a TMH model within the broader Head to Health (H2H) initiative (initially branded Head to Help). H2H was developed as a collaboration between the Australian government and the Primary Health Networks (PHNs) [15]. PHNs (31 across Australia) are regional health planning organizations that work with general practitioners as well as with other primary and secondary care providers and hospitals to facilitate better health outcomes for communities [16].

The H2H program was a free centralized intake service and had four key elements: Referral, Assessment, Treatment, and Follow-up. Referral was via a toll-free number. The referrer could speak to a mental health professional [15] who assessed the client’s needs using a specifically designed screening tool [17] and referred them to the appropriate level of care. Using a logic algorithm, the tool recommends the level of care needed for the patient according to the stepped care approach [17]. Accordingly, clients were referred to one of five different levels of care depending on the severity of their illness. Level 1 care recommended self-management for clients with minimal problems and Level 5 involved referral to acute specialist or community mental health services for clients with very severe problems. Clients who were assessed as requiring care at Levels 2–4 were then transferred to psychologists or other clinicians based at program hubs or satellite centers instituted specifically for the H2H program. See Figure 1.

The opinions of those involved in the implementation of a program are considered to be vital in determining its usefulness [18,19]. Views of the clinicians of community mental health programs, including e-mental health services, have been reported previously [20,21,22]. Understanding service user perspectives is also crucial to improving the experience of care [23]. Even those with severe mental illness are more than capable of describing preferences for both the manner in which treatments are delivered and regarding specific treatment approaches [24,25]. This study is part of a larger evaluation of the Gippsland H2H program and aims to examine the views and experiences of clinicians and users of TMH in Gippsland during the COVID-19 pandemic, with a focus on identifying what worked and did not, as well as ways to improve the service.

## 2. Materials and Methods

### 2.1. Setting

The Gippsland region is located in the southeastern corner of the state of Victoria and covers 41,600 square kilometers, with a population of over 270,000 [26]. It extends from about 100 km southeast of Melbourne to the east and the New South Wales border to the west. The median personal, family, and household income levels for Gippsland are below those of both the state and national rates [26].

Community mental health services in Gippsland are provided by the public area mental health service through its five branches. A limited range of psychosocial supports are delivered by a small number of non-government organizations. There are also a number of private mental health practitioners. The region’s geography, topography, and dispersed population, together with the acute demand caused by the COVID-19 pandemic, have compromised the regional clinical service’s capacity to provide and maintain support for people with mental illness. Gippsland PHN adopted a ‘hub and satellite’ model. Hubs and satellite centers were established based on the prevalence of mental health problems in the population and the availability of appropriate facilities and infrastructure. These hubs and satellite centers were staffed by psychologists, mental health nurses, social workers, alcohol and drug workers, and peer support workers, thereby offering multidisciplinary care accessible in person (outside of lockdowns) and via TMH at all times. See Figure 2. Satellite centers mostly complemented the hubs and had fewer staff. Hubs and satellites were located within the premises of general practices or community health centers.

Due to COVID lockdowns and the Victorian state-wide response, TMH was added to the existing service system. A state-wide evaluation of the H2H program highlighted that “remote contact (comprising telephone, video and internet-based) formed the majority (61 per cent) of all service contact modalities” across Victoria [27]. However, in the Gippsland region—the focus of this study—TMH was typically combined with face-to-face contact [27]. This study therefore provides some important insights into the use of TMH.

### 2.2. Study Design

This was a qualitative study underpinned by qualitative description that aimed to present a rich, straight description of participants’ views on the topic [28]. Accordingly, we strived to stay close to the data by avoiding excessive interpretation [29]. Qualitative description presents rich information grounded in cultural and environmental contexts. This makes research using this approach understandable to clinicians and administrators who are responsible for making the necessary changes to service delivery [30]. Qualitative description is therefore considered advantageous for obtaining information that is particularly useful to clinicians and policymakers.

### 2.3. Participant Recruitment

Clinicians were recruited using email invitations, which were distributed to GPs and mental health clinician networks. The email included a hyperlink where prospective participants could view the explanatory statement and consent form. Invitations to GPs in the region were also sent in newsletters. Clinicians interested in participating in the research returned signed consent forms to the researchers by email. Service users were recruited through a Facebook advertisement, word of mouth, and/or print advertisement at the H2H hub and satellite centers by staff who were not involved in the research. Service users who returned signed consent forms were contacted by the researchers to schedule interviews.

### 2.4. Data Collection

Data from the clinicians were collected using an anonymous online survey instrument developed specifically for the study. The instrument was built on the Qualtrics platform [31] and included demographic details, followed by 10 open-ended questions. See Appendix A.

Data from the service users were collected using one-to-one, single-session Zoom interviews, which were recorded. Audio files were transcribed using Otter software [32], which were then reviewed and corrected by the authors. The interview schedule included questions focused on the ease of accessing and entering the service, including intake and risk assessment processes, users’ views on the care received and if it met their needs. The average interview duration was 33 min. The data were collected during the month of February 2022.

### 2.5. Data Analysis

Analysis commenced with collating responses from each question. This was followed by thematic analysis [33]. Accordingly, chunks of data were inductively assigned codes. Two authors independently developed codes from the data, which were later compared and finalized. Codes were refined by ensuring that they accurately represented the data. Following this, similar codes were grouped into broader topics called categories [34]. Service user data related to experiences with TMH are included in this paper. Once all data were tentatively coded, all authors reviewed the codes and categories and made suggestions. Service user data related to other interview questions have been included in another report that is yet to be published. Categories, codes, and representative quotes are presented in the Appendix A, and the categories are briefly described in the text.

## 3. Results

Data were collected from a total of 66 participants (survey responses from 47 clinicians and interviews with 19 service users). The demographic characteristics of the clinicians and users are given in Table 1 and Table 2, respectively. As illustrated in Table 1, the majority of clinicians were female (74.5%), reported that they had worked in Gippsland’s mental health and wellbeing service sector for 10 years or more (55.3%), and had provided mental health care in Gippsland (72.3%). A little over a third (36.2%) of the clinicians were based in Latrobe City, and 42.6% were based in the two hub catchment areas (i.e., Baw Baw and Wellington Shires). As shown in Table 2, the mean age of the service user participants was 47 years. The majority identified as female (73.7%) and heterosexual (89.5%), had an Australian cultural background (84%), were employed full-time (31.6%), had attained an educational level beyond year 12 (68.4%), lived in their own home (73.7%), had no carer (94.7%), and reported having depression (54.9%).

Six categories emerged from the data on the use of TMH during the COVID-19 pandemic in Gippsland. They were: ‘Conditions where use of tele-mental health is appropriate’, ‘Conditions where tele-mental health may not be useful’, ‘Advantages of tele-mental health’, ‘Challenges in using tele-mental health’, ‘Client outcomes with tele-mental health’, and ‘Recommendations for future use’. These categories are discussed below. The code tree and representative quotes are given in Appendix A.

### 3.1. Conditions Where Use of TMH Is Appropriate

As the name suggests, this category relates to conditions and situations where the use of TMH was found to be appropriate and useful. Both clinicians and users indicated that TMH rendered mental health services much more accessible for those who resided in geographically isolated areas and was found to be particularly useful in enabling access during the lockdown period. Clinicians stated that given the general paucity of mental health services and the high unmet need in the region, TMH was very welcome. They found it to be effective for young adults and those who were able to independently set up and use technology. Clinicians also indicated that the model worked well when there was an organizational commitment to using the technology and that TMH was useful when clients had anxiety, were working on maintaining recovery, and with whom there was a pre-existing therapeutic relationship. Clinicians also used TMH to stay connected with their clients. Service users with access to the technology and internet also found it useful. Aboriginal service users in particular, who typically experience a lack of appropriate services, had high praise for this mode of service delivery.

### 3.2. Conditions Where TMH May Not Be Useful

There were certain conditions where TMH was found to be unhelpful. Clinicians felt that it was not conducive as a service for individuals who could not afford a phone or internet or for those who belonged to culturally and linguistically diverse (CALD) communities owing to language barriers. However, a service user from a CALD background who shared this sentiment at the outset due to her accent, less-than-optimal vocabulary, and a general sense of unease to open up to a stranger, was pleasantly surprised by her experience owing to a compassionate and encouraging clinician. Clinicians felt that reputational issues related to TMH, such as it being a ‘second-rate’ service, were also a barrier to its use by clients from certain smaller rural towns. They also believed that TMH was not helpful for children who required hands-on therapy activities or older clients who were not familiar with using technology. Finally, clinicians stated that TMH was not appropriate when working with clients in distress who needed immediate human connection and for those who were at risk of self-harm.

### 3.3. Advantages of TMH

TMH was found to have several advantages over face-to-face service delivery. For instance, it clearly improved accessibility in that it could be delivered in the safety and familiarity of the home, and clients did not have to travel long distances to see a specialist. While one service user credited TMH for giving them the opportunity to access a psychologist without delay, another was pleased that TMH was available because it gave them privacy, which can be a significant issue in rural and remote locations. Referring clinicians also found it useful to ask questions and receive direct feedback. Clinicians stressed the advantage of being able to phone clients even when the latter had forgotten the appointment. Service users, on the other hand, were pleased that they could answer a phone call wherever they were, even when they had forgotten the appointment or were unable to attend in person. Clinicians preferred phone consultations for clients who did not have access to the internet or to other video conferencing (VC) devices. Using the telephone was also beneficial to mental health clinicians when they had to work from home due to COVID-19 restrictions. One service user also stated that they were more comfortable with phone consultations and that they found video conferencing quite awkward. Clinicians found VC useful when they were required to have an idea of the client’s surroundings, especially when emergency services needed to be dispatched.

### 3.4. Challenges of Using TMH

Despite there being advantages to using TMH, there were also several challenges. Service users complained about frequent breakdowns of the videoconference links, which resulted in missed appointments. Appointments also did not materialize when clients were not sent sufficient information on how to use technology for TMH prior to sessions. Clinicians reported that several clients required assistance with using TMH. Not only clients but also clinicians took time to adjust in the absence of face-to-face interactions. Therefore, when using TMH, extra time was needed to allow for overcoming technical difficulties. Clinicians continue to learn ways of building rapport with clients over the phone or videoconference.

In more remote areas, where internet connections were unreliable, clients were forced to use the telephone. However, problems such as lag times and flat batteries added to the disruption of interactions. Some clinicians indicated that clients faced difficulties finding suitable locations in their homes to have private conversations. There were also other challenges with using the phone in TMH. For example, clinicians felt that observation of client behavior and client engagement, assessing safety levels, and conducting mental state examinations, which are imperative for clinical work in mental health, were not possible over the phone. Clinicians also felt that phone consults were quite impersonal and less effective for building rapport with clients. Some service users also found phone consultations uncomfortable, while others felt alone and unsupported on the phone, preferring face-to-face interactions instead. Clinicians felt that service users were also reluctant to open up and discuss their problems on the phone, which resulted in missing information and sessions that could not be completed. A more concerning situation was when clients at risk did not answer the phone. Finally, although VC was generally the preferred mode for TMH, young people did not prefer it, and clients from low-income backgrounds could not afford it.

### 3.5. Client Outcomes with TMH

Outcomes were positive when clients were comfortable using TMH, and were generally doing well. Negative outcomes resulted when the lack of face-to-face interactions distressed clients who were in isolation. Some clients’ symptoms appeared to get exacerbated with screen and phone interaction. Service providers were concerned that in situations such as the COVID lockdowns, where people tended to stay indoors, telehealth could further serve as an impediment to accessing services. This could worsen isolation in individuals who have difficulty leaving their homes.

### 3.6. Recommendations for Future Use

Clinicians made several recommendations for the future use of TMH services. They suggested that TMH was best suited for client–practitioner interactions when there was an existing therapeutic relationship between them. It was best used as part of a ‘dual mode’ of delivery rather than a stand-alone model because it could not take the place of face-to-face interactions. Clinicians also recommended that clients needed to be screened to check if they were ‘TMH ready’ before using the model with them.

Furthermore, clinicians indicated that there needed to be communication with clients, prior to the commencement of the session, about the use of technology and how the service would work. They were also of the opinion that the use of TMH was in the development stage and that there was much to grow and improve before it became easy to access and trusted by clients. Issues such as maintaining the privacy of both the client and the practitioner needed to be ironed out. Clinicians also needed to receive training on the appropriate use of technology and ways to develop rapport when using TMH. Finally, some proposed the setting up of TMH kiosks in areas where services were few and far between and where clients who did not have access to devices could engage with TMH services in a safe and private environment.

## 4. Discussion

This study provides a broad range of information on what works and what does not when TMH is implemented alongside a public mental health service at a time of crisis, such as that encountered during the COVID-19 pandemic. This is one of the few studies where service users’ views and experiences were explored together with those of clinicians so as to provide dual perspectives of the service. The findings affirm that in situations where mental health services are typically difficult to access, a TMH service is most appropriate despite its shortcomings. This finding has been confirmed by other authors as well [3].

In jurisdictions where TMH is still in its infancy, organizational support and promotion of the technology are needed to establish its routine use. Other reports have alluded to this finding by suggesting that regulatory, licensure, and clinical issues must be addressed prior to offering TMH services [35]. In the H2H program, TMH services were delivered by phone and VC. Both modes have advantages and disadvantages. Whilst telephones are more convenient and widely available, as discussed earlier, there are major shortcomings with their use. The effectiveness of telephone-based interventions also needs further evaluation [36]. On the other hand, the greatest challenge with using VC was its lack of universal availability as well as clients’ lack of knowledge and ability to use it. Service users also needed to be screened to ensure that they had access to good internet bandwidth and appropriate devices and the ability to use it. This is a well-known challenge of using TMH [5], particularly among older people [9]. Some authors have suggested that when using TMH, there needs to be someone to manage technological issues [37]. Furthermore, in geographically isolated regions with poor internet connectivity, TMH kiosks were suggested as a solution, which could provide the privacy and safety needed for clients. However, in such instances, factors such as financial sustainability would need to be considered [38].

Clinicians felt that TMH posed challenges to developing a therapeutic alliance with their clients. This alliance is fundamental to the success of any face-to-face psychological therapy. Previous studies have also highlighted this challenge [35,39] and have suggested that it is possible to develop this alliance [40,41], although it may have unique and yet unconfirmed characteristics [42], particularly when working with children [43]. The finding that young people, despite being comfortable using technology, did not prefer videoconferencing is perhaps due to their desire to maintain confidentiality. This resembles the findings of a systematic review which showed that online self-help platforms were the most frequently used modality by young people [44].

The clinicians in this study indicated that TMH was most useful when used in combination with face-to-face interactions and for the continuity of care and maintenance of recovery. The role and usefulness of TMH as an adjunct to face-to-face care have also been reported by other authors [45,46,47].

In such ‘dual-mode’ services, initial consultations that focus on developing rapport and setting the care plan can be undertaken in face-to-face interactions, whereas follow-up care can be conducted via TMH. This combination of use has the potential to smoothly build rapport between the client and clinician before interactions are continued by TMH. In this study, whether or not recovery-oriented care by TMH described by the participants simply involved a routine follow-up is not known. Nonetheless, TMH-based recovery interventions are still in the development phase [48].

Whilst there are explicable challenges when utilizing TMH for CALD communities, these can be eminently addressed when a clinician has a compassionate and positive attitude and has received appropriate training [49]. As other reports have indicated, this could be achieved with relevant training in effective communication strategies and culturally sensitive care [50]. Some have proposed that a liaison person could facilitate interactions with persons from a CALD background [51]. Interestingly, a report from the UK has shown how TMH was a preferred option for black and other minority communities due to the community stigma attached to accessing traditional mental health services [52]. Walker and colleagues reported that when used with Aboriginal young people, TMH resulted in good outcomes for wellbeing and cultural continuity, although only about 63% of families had internet connectivity in their homes [53]. Furthermore, the finding that both clients and clinicians found it difficult to navigate the system and, hence, lost time, is significant. In the United States, where TMH is a routine part of mental health services, service users and clinicians are given guidelines on how to use the technology [35]. Perhaps that will be replicated in Victoria once TMH services are rendered as a part of routine care.

This study also highlighted the potentially harmful effects of TMH. Although the H2H model was not designed as an acute service, it was also accessed by clients who were in distress and at risk of self-harm. Such situations were challenging for clinicians and users because what the client required at this time was immediate person-to-person interaction. Referring such clients to the Emergency Department was not considered optimal. This finding concurs with that of a previous report [5] and acknowledges that such services need to have appropriate protocols in place to manage situations where individuals who access the system are in a state of distress or require emergency care.

Clinicians using TMH were unable to assuage the significant distress experienced by service users who were in COVID-19-induced isolation. There were also instances where clients’ symptoms were exacerbated by screen/phone interaction perhaps as a consequence of not having prior face-to-face social contact for a prolonged period of time. In addition, there were concerns that using TMH could worsen the wellbeing of individuals with little or no social connections and who had an existing difficulty leaving their homes.

This study had some limitations. Clinicians who participated in the survey might not have had the inclination to expand on their views, preferring rather to make a short point, and thereby forfeiting some of the detail in their responses. However, a survey was considered a better option for otherwise time-poor clinicians.

## 5. Conclusions

This study reported on what worked and did not when TMH was implemented alongside a public mental health service at a time of crisis, such as that encountered during the COVID-19 pandemic. At a time where mental health services were typically difficult to access, a TMH service was most welcome despite its shortcomings. These findings have implications for policy and practice.

## Figures and Tables

**Figure 1 ijerph-20-05870-f001:**
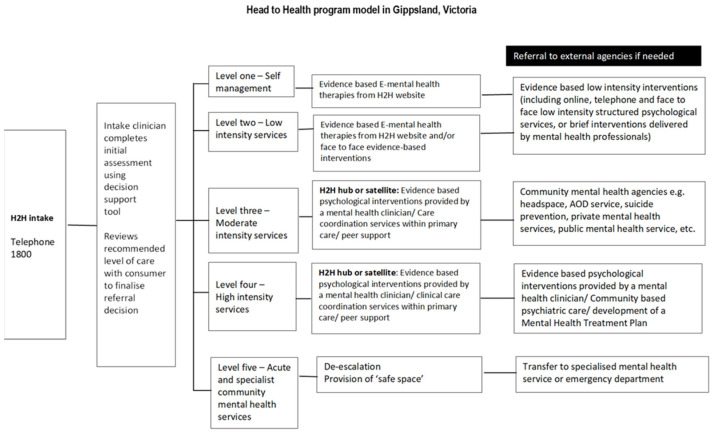
Head to Health program model in Gippsland, Victoria.

**Figure 2 ijerph-20-05870-f002:**
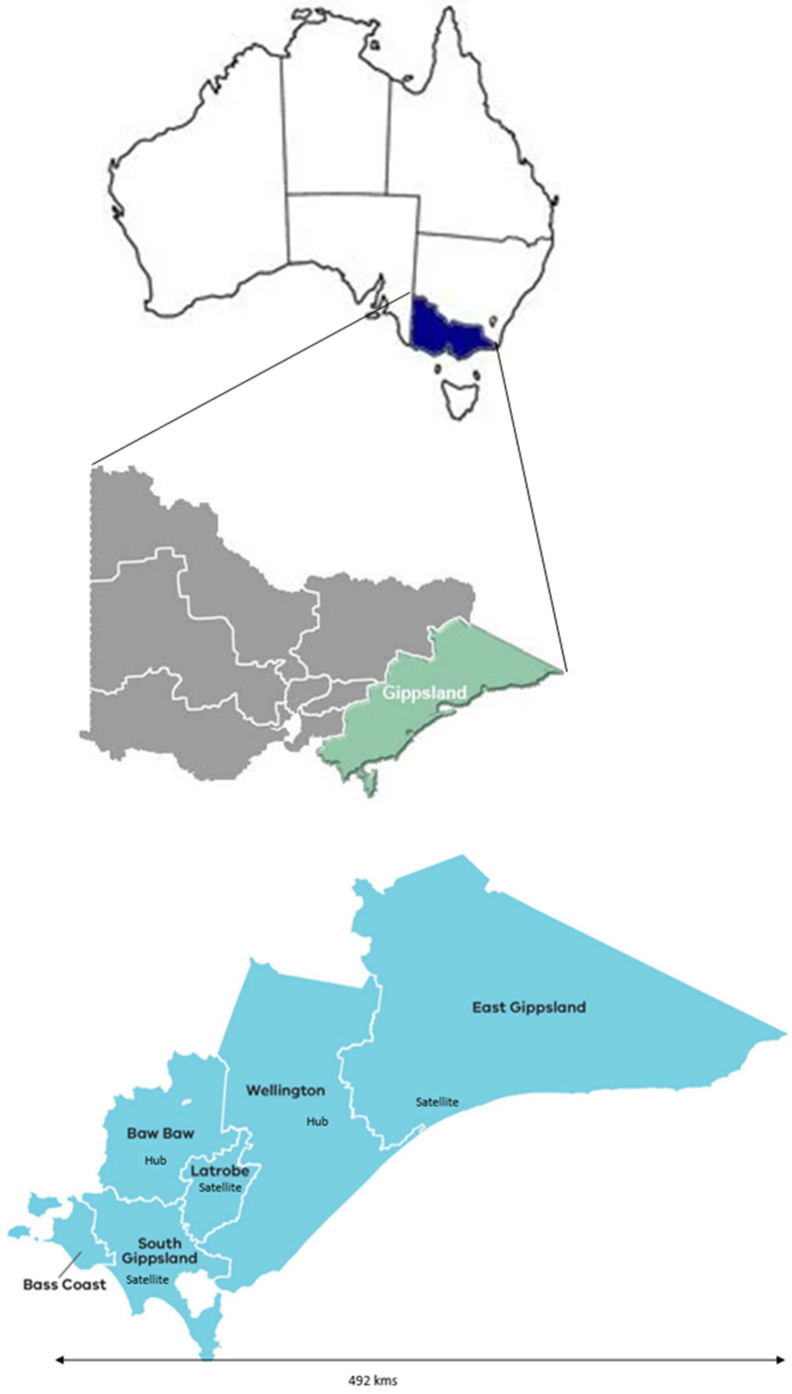
Geographical area covered by the Head to Health program in Gippsland showing local government areas and H2H hubs and satellites.

**Table 1 ijerph-20-05870-t001:** Demographic characteristics of service provider participants (N = 47).

Characteristic	Number (%)	Characteristic	Number (%)
**Gender**		**Age groups (years)**	
Women	35 (74.5)	20–29	8 (17.0)
Men	12 (25.5)	30–39	9 (19.1)
**Primary professional roles (health professional)**		40–49	11 (23.4)
General practitioner	7 (14.9)	50–59	9 (19.1)
Other medical	1 (2.1)	60 and older	10 (21.3)
Mental health nurse	3 (6.4)		
Occupational therapist	1 (2.1)	**Mental health service experience in Gippsland**	
Psychologist	2 (4.3)	<1 year	3 (6.4)
Social worker	4 (8.5)	1–5 years	14 (29.8)
Mental health support worker	7 (14.9)	5–10 years	4 (8.5)
Psychosocial support worker	1 (2.1)	>10 years	26 (55.3)
Alcohol and other drugs service worker	1 (2.1)	**Local government area**	
Emergency department worker	1 (2.1)	Baw Baw Shire	9 (19.1)
		East Gippsland Shire	5 (10.6)
**Primary professional roles (non-health practitioner)**		Latrobe City	17 (36.2)
Department of Families, Fairness and Housing worker	2 (4.2)	South Gippsland Shire	4 (8.5)
Family violence service worker	2 (4.3)	Wellington Shire	11 (23.4)
Manager/administrator	7 (14.9)	Non-Gippsland LGA	1 (2.1)
Researcher/educator	5 (10.6)	**Location of practice in LGA * within Hub**	
Volunteer	1 (2.1)	Yes	20 (42.6)
Other (not specified)	2 (4.2)	No	27 (57.4)
**Identify as working in the mental health sector**			
Yes	34 (72.3)		
No	13 (27.7)		

* LGA = Local government area.

**Table 2 ijerph-20-05870-t002:** Demographic characteristics of service user participants (N = 19).

Service User Characteristics	N (%)	Service User Characteristics	N (%)
**Gender**		**Housing**	
Woman	14 (73.7)	Own home	14 (73.7)
Man	5 (26.3%)	Public rental	3 (15.8)
**Age (years)**		Private rental	1 (5.3)
Mean	47 years	Emergency housing	1 (5.3)
Range	22–77 years	**Living arrangement**	
**Cultural background**		Alone	5 (26.3)
Aboriginal	2 (10.5)	With partner only	6 (31.6)
Australian	16 (84.2)	With partner and children	3 (15.8)
Spanish	1 (5.3)	With children only	4 (21.1)
**Sexuality**		Did not disclose	1 (5.3)
Heterosexual	17 (89.5)	**Carer status**	
LGBTQIA+	1 (5.3)	Has a carer	1 (5.3)
Did not disclose	1 (5.3)	Has no carer	18 (94.7)
**Employment status**		Care for another person	6 (31.6)
Self-employed	2 (10.5)	**Past mental health problems**	
Unemployed	3 (15.8)	Stress	2 (10.5)
Retired	3 (15.8)	Anxiety	6 (31.6)
Employed full-time	6 (31.6)	Depression	11 (57.9)
Full-time carer	1 (5.3)	Borderline personality disorder	4 (21.1)
Studying and working	1 (5.3)	Grief	1 (5.3)
Disability pension	3 (15.8)	Post-traumatic stress disorder	5 (26.3)
**Highest educational attainment**		Trauma	3 (15.8)
Year 8	2 (10.5)	Social phobia	1 (5.3)
Year 11	2 (10.5)	Bipolar	2 (10.5)
Year 12	2 (10.5)	Psychosis	1 (5.3)
Certificate IV	1 (5.3)	Dissociative identity disorder	1 (5.3)
Diploma	7 (36.8)		
Bachelor’s degree	3 (15.8)		
Graduate diploma	1 (5.3)		
Master’s degree	1 (5.3)		

## Data Availability

The data are contained within the article and Appendix A.

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
