# Peer review of "Clinicians’ and Users’ Views and Experiences of a Tele-Mental Health Service Implemented Alongside the Public Mental Health System during the COVID-19 Pandemic"

_ijerph, 2023, doi:10.3390/ijerph20105870_

Round 1
Reviewer 1 Report
It's a well-conceived qualitative study addressing the issue of increased use of telehealth services during the pandemic in the domain of mental health. The Authors clearly presented the local context of the application of the telemental health solution. The process of data collection is transparently described. The results of the thematic analysis are accompanied by supplementary material providing a deeper look into the way data were coded and categorized. I believe the paper is feasible for publishing as it is.
Reviewer 2 Report
Comments and suggestions
It was my pleasure to review this manuscript dealing with clinicians’ and users’ views and experiences of a tele-mental health service implemented alongside the public mental health system during the COVID-19 Pandemic. This manuscript aimed to understand the views and experiences of clinicians and service users of the telemental health service in the Gippsland region of Victoria, during the COVID-19 pandemic. A cross-sectional study through semi-structured interviews was conducted, and a total of 66 participants including 47 clinician surveys and 19 service user interviews were used for analysis. In brief, this manuscript has a well-structured description. I found the topic quite interesting, and it is worth investigating. I suggest this manuscript needs to be undergone extensive English revisions. Because many minor mistakes exist and should be corrected. But with the sole objective of improving the quality of the manuscript, I will allow myself to make a few comments.
Introduction part
1. The introduction has a well-structured description; meanwhile, the contents included some latest references during the COVID-19 pandemic. Hence, it is enriched with informative references drawing from the current rich international references.
Materials and Methods
1. The manuscript did not mention the study sampling of this study. It only mentioned that clinicians interested in participating in the research returned signed consent forms to the researchers by email. Service users were recruited through a Facebook advertisement, word of mouth, and/or print advertisement at the H2H hub and satellite centers by staff who were not involved in the research. Was it done randomly or just for convenience?
2. It is not specified how the sample size was calculated to be representative. A margin of error of 5% and a confidence interval of 95% is normally accepted. The number of individuals that made up the study population was not expressed, nor was the minimum number of the necessary sample meeting the criteria I listed above.
3. The manuscript mentioned that data from clinicians were collected using an anonymous online survey instrument and data from service users was collected using one-to-one, single-session, zoom interviews which were recorded. The manuscript didn’t mention when the study was conducted and what the response rate of this study was. This information needs to be added to the revised manuscript..
Results
4. Page 8, lines 259-261:
You mentioned that “Similarly, negative outcomes ensued when the availability of TMH services did not challenge clients with otherwise little social interaction, to leave their homes”. I suggest you need to address this sentence more clearly to let readers thoroughly realize your expression or you can add some examples to deliver your meaning.
Discussion
The results may not be representative and exist as a question of external validity. Convenience samples are quite prone to research bias. Since the researcher draws the sample based on convenience and not equal probability, convenience samples never result in a statistically balanced selection of the population. This leads to sampling bias. This part should mention in the study limitation.

Reviewer 3 Report
Coming from the developing world it is always surprising the number of interventions that seem cost effective yet not scalable in other regions of the world. I congratulate the research team for showing us how we can address access and geographical barriers.
Reviewer 4 Report
This study looked into what worked and did not when TMH was implemented alongside a public mental health service during pandemic. It is very interesting and well written study.
My only remarks is related to the methodology part. I suggest adding little more information on the coding process (how many people were involved in the process, did they agree on categories right away,...). And I believe that you could reflect on this coding process in the limitation section because it is highly subjective, and it has to be interpreted with caution.
Overall, I congratulate you on writing this very good paper and in my opinion, it is worthy publishing.
Reviewer 5 Report
Dear authors, thank you for the opportunity to get acquainted with the results of your current research. Undoubtedly, evaluating the effectiveness of telemedicine technologies is an important issue for healthcare. At the same time, the manuscript has a number of shortcomings and incomplete presentation of scientific information:
1. The introduction does not contain a clear scientific goal of the study.
2. In the Materials and Methods section, the authors do not present the specific questions that were asked to the respondents, nor do they provide a rationale for choosing these questions.
3. The Materials and Methods section does not specify in detail how the content analysis of the data was carried out, how the answers were combined into blocks?
4. The authors did not indicate why the qualitative research paradigm was chosen, and why it was not supplemented by quantitative research? Was there an analysis of objective data on the effectiveness of this program? And their comparison with real results
5. The results of the study are presented descriptively, only data relative to the study sample are quantitatively presented. I would like to see the frequency of respondents' answers on the stated topics? Which of the above problems are most relevant?
6. The discussion of the results does not contain conclusions on how to improve the telemedicine program based on the results of the study.
7. Limitations of the study are written very briefly. Although the authors have applied only a qualitative paradigm, it is not specified what limitations it contains. Research prospects are not specified.
8. The conclusion is not specific. It is not disclosed what exactly worked and what did not. What are the ways to optimize?
In connection with the above, the article needs significant revision.
Best regards, reviewer.
Round 2
Reviewer 5 Report
Dear authors, comments have been removed.
The article can be recommended for publication.
Best regsrds, the reviewer